# Effects of Dietary Fatty Acids from Different Sources on Growth Performance, Meat Quality, Muscle Fatty Acid Deposition, and Antioxidant Capacity in Broilers

**DOI:** 10.3390/ani10030508

**Published:** 2020-03-19

**Authors:** Shenfei Long, Sujie Liu, Di Wu, Shad Mahfuz, Xiangshu Piao

**Affiliations:** State Key laboratory of Animal Nutrition, College of Animal Science and Technology, China Agricultural University, Beijing 100193, China; longshenfei@cau.edu.cn (S.L.); heiluobo12300@163.com (S.L.); superwudee@163.com (D.W.); shadmahfuz@yahoo.com (S.M.)

**Keywords:** broilers, fatty acid, fish oil, linseed oil, microalgae, performance

## Abstract

**Simple Summary:**

The findings in the current study reveal that dietary fish oil or a combination of linseed oil and microalgae could be effective in improving growth performance, carcass traits, muscle fatty acid deposition, and antioxidant capacity in broilers compared with traditional soybean oil in broilers.

**Abstract:**

This study aimed to investigate the efficiency of dietary fatty acids from various sources on growth performance, meat quality, muscle fatty acid deposition and antioxidant capacity in broilers. 126 Arbor Acres broilers (1 d-old, initial body weight of 45.5 ± 0.72 g) were randomly assigned to three treatments with seven cages per treatment and six broilers per cage. The dietary treatments included: (1) corn–soybean meal basal diet containing 3% soybean oil (control diet, CTL); (2) basal diet + 1% microalgae + 1% linseed oil + 1% soybean oil (ML); (3) basal diet + 2% fish oil + 1% soybean oil (FS). The trial consisted of phase 1 (day 1 to 21) and 2 (day 22 to 42). Compared with CTL, broilers fed ML or FS diet showed improved (*p* < 0.05) average daily gain in phase 1, 2, and overall (day 1 to 42), as well as a decreased (*p* < 0.05) feed conversion ratio in phase 1 and overall. On day 42, broilers supplemented with FS diet showed increased (*p* ≤ 0.05) the relative weights of pancreas and liver, as well as higher (*p* < 0.05) redness value in breast and thigh muscle compared with CTL. Broilers offered ML or FS diet had lower (*p* < 0.05) the relative weight of abdominal fat and total serum cholesterol content in phase 1, and increased (*p* < 0.05) contents of serum glucose, n-3 polyunsaturated fatty acids (PUFA), eicosacagetaenoic acid, docosahexaenoic acid, glutathione peroxidase, superoxide dismutase and total antioxidant capacity, as well as lower (*p* < 0.05) concentrations of malondialdehyde, n-6 PUFA, and n-6/n-3 PUFA ratio in breast and thigh muscle compared with CTL. This research indicates that diets supplemented with fish oil or a combination of microalgae and linseed oil experience improved performance, antioxidant capacities and n-3 PUFA profile in muscle of broilers compared with traditional soybean oil supplemented diets

## 1. Introduction

Meat harvested from animals plays an important role in human diets due to its fatty acids, protein, mineral, and vitamin contents. In recent decades, the consumption of chicken meat has increased and become widely popular around the world. Modern poultry production focuses on improving the performance and health status of broilers, and producing poultry meat that is nutritious for consumers. Studies have shown that high concentrations of n-3 polyunsaturated fatty acids (PUFA), such as docosahexaenoic acid (DHA, C22:6n-3), eicosacagetaenoic acid (EPA, C20:5n-3), and alpha-linolenic acid (ALA, C18:3n-3)—as well as an increased ratio of PUFA to saturated fatty acid ratio (PUFA/SFA ratio) and n-3/n-6 PUFA ratio—are necessary and beneficial for broilers and human health [1]. However, the intake of n-3 PUFA in the average human diet is low and the conversion of ALA to EPA and DHA in the human body is also low [2]. Therefore, scientists have recently shown more interest in producing poultry meat products enriched with n-3 PUFA for human consumption [3].

The fatty acid composition in the breast and thigh muscles of animals is highly related to the fatty acids in the diet fed to animals [4], so providing animals with diets enriched in n-3 PUFA can be the most practical way to manipulate the n-3 PUFA profiles of meat [5]. Previous studies have shown that the addition of linseed oil (LO), microalgae (MA), fish oil (FO), or a mixture of these in diets fed to poultry can be effective in increasing fatty acids—namely n-3 PUFA [6,7]. The FO is enriched with EPA and DHA, which can improve performance, anti-inflammatory functions and n-3 PUFA deposition in chicken meat [8]. However, adding high levels of FO might cause adverse flavor and color changes to chicken meat [6]. The LO is enriched with ALA, which serves as the metabolic precursor of DHA and EPA [9]. Studies show that dietary LO supplementation can also enhance the ALA, EPA and DHA profiles in the muscle of broilers [7]. The MA is enriched with DHA, which has been shown to effectively improve n-3 PUFA concentrations in the muscle of broilers [10]. Moreover, MA can effectively improve the oxidative stability in chicken muscle compared with FO [11]. However, this is up for debate, as another study reported no difference in the oxidative stability of chicken meat between MA and FO-fed diets [12]. Although MA, LO, and FO have all demonstrated the ability to be excellent sources of different fatty acids, especially n-3 PUFA, few studies have reported the mixed effects of microalgae, linseed oil, and fish oil compared with traditional soybean oil (SO) in poultry diets. Based on these previous findings above, we hypothesize that dietary MA and LO supplementation might be a novel combination for improving health status and muscle fatty acid profile in broilers in comparison with FO or traditional SO.

Therefore, our study focuses on better estimating and understanding the effects of FO or a combination of LO and MA supplementation on growth performance, carcass traits, muscle fatty acid deposition, and antioxidant capacity in broilers compared with traditional SO-supplemented diets.

## 2. Materials and Methods

The experimental protocols used in this experiment were approved by the Laboratory Animal Welfare and Animal Experimental Ethical Inspection Committee of China Agricultural University (Beijing, China; No. AW09089103-1). The trial was carried out at FengNing Research Unit of China Agricultural University (Academician Workstation in Chengdejiuyun Agricultural and Livestock Co., Ltd).

### 2.1. Experimental Products

The SO, FO, and LO used in this experiment were supplied by Beijing TongliXingke Agricultural Science and Technology Co., Ltd. (Beijing, China). Alltech Inc. (Nicholasville, KY, USA) offered the MA (*Schizochytriumlimacinum* CCAP 4087/2) product (in powder form), which has been proved effective in enriching n-3 PUFA deposition in broilers [10]. All the fatty acid compositions in these oils are presented in Table 1.

### 2.2. Experimental Animals, Management, and Design

A total of 126 Arbor Acres broilers (1 d-old, initial body weight of 45.5 ± 0.72 g, offered by Beijing Arbor Acres Poultry Breeding Co., Ltd. (Beijing, China)) were randomly assigned to 3 treatments (7 cages in each treatment, 6 broilers per cage). The treatment included a control diet (CTL, corn-soybean meal basal diet + 3% SO, fed in mash form), an ML diet (basal diet + 1% MA + 1% LO + 1% SO), and an FS diet (basal diet + 2% FO + 1% SO). The trial consisted of two phases with phase 1 (day 1 to 21) and phase 2 (day 22 to 42). Nutrient levels in all formulated diets for each phase met or exceeded the nutrient requirements recommended in NRC (1994, Table 2).

All broilers were kept in an environmentally controlled room under continuous illumination (10–20 Lux) and supplied feed and water ad libitum. The room temperature was set and controlled at about 33 °C for the first three days and decreased by 3 °C per week until reaching 24 °C. On day 7 and 28, broilers were vaccinated with Newcastle disease vaccine. On day 14 and 21, all broilers were vaccinated with an inactivated infectious bursal disease vaccine.

### 2.3. Sampling and Processing

All diet samples were ground to pass through a 0.45-mm sieve and samples were prepared for analysis of crude protein (CP, AOAC 2001.11-2005), and dry matter (DM, AOAC 2001.12-2005). An automatic isoperibolic oxygen bomb calorimeter (Parr 1281, Automatic Energy Analyzer; Moline, IL, USA) was used to determine the gross energy (GE) content of the diets.

Broilers and feeders were weighed on day 21 and 42 after fasting for 12 h to calculate average daily gain (ADG), average daily feed intake (ADFI), and feed conversion ratio (FCR) of the broilers.

### 2.4. Serum Parameters Measurement

On day 21 and 42, blood samples (about 8 mL, n = 7) were collected from one broiler close to average body weight in each cage via jugular vein puncture into a 10-mL anticoagulant-free Vacutainer tube (Becton Dickinson Vacutainer Systems, Franklin Lakes, NJ, USA). After stewing for about three hours, blood samples were centrifuged at 3000× *g* for 10 min at 4 °C to recover serum, which was then stored at −80 °C until further analysis. The contents of low-density lipoprotein cholesterol (LDL-C), triglyceride (TG), high-density lipoprotein cholesterol (HDL-C), total cholesterol (TC), globulin, albumin, glucose, total protein (TP), and urea nitrogen (UN) in serum were tested (colorimetric method) following the corresponding reagent kit manufacturer instructions (Zhongsheng Biochemical Co., Ltd., Beijing, China) via an automatic biochemical analyzer machine (RA-1000, Bayer Corp., Tarrytown, NY, USA). The concentration of malondialdehyde (MDA) was determined by Thiobarbituric Acid Reactive Substance (TBARS) Assay Kit (Cell Biolabs, Inc., San Diego, CA, USA) following the manufacturer’s protocols. The activities of superoxide dismutase (SOD), catalase (CAT), total antioxidant capacity (T-AOC), and glutathione peroxidase (GSH-Px) in serum were determined by spectrophotometric methods using a spectrophotometer (LengGuang SFZ1606017568), according to the SOD, CAT, T-AOC and GSH-Px kits’ manufacturer instructions (The Nanjing Jiancheng Bioengineering Institute, Nanjing, China).

### 2.5. Meat Quality and Carcass Traits Measurement

On day 21 and 42, one broiler per cage (near the average body weight, n = 7) was chosen to evaluate carcass traits and meat quality. The muscle color (lightness, redness, and yellowness values) in the left breast muscle and thigh muscle were measured from 3 orientations (middle, medial, and lateral) using a Chromameter (CR-410, Konica Minolta, Tokyo, Japan) according to the method by Elokil et al. [13]. The plastic bag method was utilized to determine drip loss in the breast and thigh muscles. On day 42, the relative weights of heart, spleen, liver, abdominal fat, pancreas, and fabricius of broilers were calculated using the following formula: weight of organ/final body weight × 100% = Relative weight of organs (%BW).

### 2.6. Fatty Acid Profile Testing

On day 42, after euthanasia or harvesting, one broiler per cage (near the average body weight, n =7) was chosen to measure the fatty acid profile in the left breast and left thigh muscle. The homogenized skinless freeze-dried breast and thigh muscle along with the experimental milled diets utilized in phase 2 were defrosted. A freeze dryer was used to lyophilize about 10 g diet and 20 g muscle for 60 h. A solvent mixture of chloroform and methanol (2:1, vol/vol) was used to extract total lipids from the homogenized muscle tissue. Gas chromatography (GC, 6890 series, Agilent Technologies, Wilmington, DE, USA) was used to determine the fatty acid profile in diets and muscles following the procedure used by Long et al. [10]. The calculation for n-6 PUFA, n-3 PUFA, the n-6/n-3 PUFA ratio, SFA, the PUFA/SFA ratio, and the monounsaturated fatty acids (MUFA) followed the same procedure outlined in Long et al. [10].

### 2.7. Statistical Analysis

Data were analyzed using the MIXED procedures of SAS (version 9.2, 2008; Inst., Inc., Cary, NC, USA). The dietary treatment served as the fixed effect, and body weight was the random effect. The individual cage was classified as the experimental unit in terms of performance data (body weight gain, ADG, ADFI and FCR). Individual broilers selected in each cage were considered as the experimental unit for all other indices. Student–Neuman–Keul’s Multiple Range Test was used to separate the statistical differences among all the treatments. Least squares means and SEM were used to express the results. The significance was defined at *p* ≤ 0.05, and a trend for significance was determined at 0.05 < *p* ≤ 0.10.

## 3. Results

### 3.1. Fatty Acids Profile in Diets

As presented in Table 3, the, C18:1n-9, C20:0, C16:0, C18:0, and C18:2n-6 were the major fatty acids present in the diets. With the addition of MA and LO, concentrations of n-3 PUFA and DHA increased by approximately 2.3%, whereas the n-6/n-3 PUFA ratio approximately decreased 83% in phase 1 and 94% in phase 2. With the addition of FO, concentrations of n-3 PUFA, EPA and DHA showed an approximate increase of 0.6%; whereas the n-6/n-3 PUFA ratio decreased by approximately 78% in phase 1 and 84% phase 2. 

### 3.2. Growth Performance

Compared with CTL, broilers fed the ML and FS diet showed improved (*p* < 0.05) average daily gain in phase 1 (day 1 to 21), 2 (day 22 to 42), and overall (day 1 to 42), and decreased (*p* < 0.05) feed conversion ratio during phase 1. Additionally, broilers offered the FS diet tended to have an increased (*p* = 0.09) ADFI in phase 1 compared with CTL (Table 4).

### 3.3. Meat Quality and Carcass Traits

Compared with CTL, broilers offered the FS diet showed lower (*p* ≤ 0.05) lightness in thigh muscle on day 21, while expressing higher (*p* ≤ 0.05) redness on day 42. Broilers fed the FS diet also had greater (*p* < 0.05) relative weights of the liver and pancreas. Additionally, broilers offered the ML or FS diet showed decreased (*p* < 0.05) the relative weight of abdominal fat. However, there were no significant effects on the thigh and breast drip loss in broilers supplemented with ML or FS diet compared with CTL (Table 5).

### 3.4. Serum Composition

Compared with CTL, broilers offered the ML or FS diet showed a lower (*p* < 0.05) content of TC during phase 1. In phase 2, broilers fed the ML or FS diet had higher (*p* < 0.05) content of glucose and tended to have lower concentrations of TC and UN in serum compared with CTL (Table 6).

### 3.5. Antioxidant Status in Muscle

Compared with CTL, broilers fed the ML or FS diets showed greater (*p* < 0.05) activities of T-AOC, SOD and GSH-Px, as well as a reduced (*p* < 0.05) concentration of MDA in the breast and thigh muscles; while broilers fed the ML diet showed an improved (*p* < 0.05) activity of CAT in the thigh muscles (Table 7).

### 3.6. Fatty Acid Composition in Muscle

In breast muscle, broilers offered the ML or FS diet had improved (*p* < 0.05) concentrations of EPA, DHA, C14:0, C17:0, and n-3 PUFA, as well as decreased (*p* < 0.05) contents of n-6 PUFA, C18:2n-6, and n-6/n-3 PUFA ratio compared with CTL. Broilers fed the FS diet showed lower (*p* < 0.05) concentrations of ALA, PUFA, and PUFA/SFA ratio, as well as a higher (*p* < 0.05) content of C20:1n-9 compared with CTL. Furthermore, broilers fed the ML diet showed an increased (*p* < 0.05) content of ALA compared with CTL, and tended to have lower concentrations of C18:1n-9 and MUFA compared with broilers offered the FS diet (Table 8).

In the thigh muscles, compared with CTL, broilers offered the ML or FS diets had increased (*p* < 0.05) concentrations of EPA, DHA, SFA, C14:0, C17:0, and n-3 PUFA, as well as decreased (*p* < 0.05) contents of MUFA, C18:1n-9, C18:2n-6, C18:3n-6, n-6 PUFA, n-6/n-3 PUFA ratio. Moreover, broilers fed the FS-supplemented diet showed higher (*p* < 0.05) concentrations of C18:0 and C20:4n-6, and tended to have a higher (*p* = 0.08) content of C20:1n-9. Broilers fed the FS-supplemented diet also had lower (*p* < 0.05) concentrations of ALA and PUFA/SFA ratio compared with the CTL. Broilers offered the ML diet had a higher (*p* < 0.05) content of ALA compared with CTL (Table 9).

## 4. Discussion

In the present study, we demonstrated that the ADG was improved in broilers fed ML-supplemented diets compared with a typical soybean meal-corn based diet. This may be partly due to the inclusion of 1% and 2% MA, which could improve ADG and decrease FCR in broilers [7,10]. This result may also be due to the beneficial effects of high level-ALA in LO and high level-DHA in MA on moderating the microflora composition in cecum and improving intestinal morphology in broilers [14]. Moreover, a mixture of MA and LO might slow the feed passage rate in the small intestine and increase nutrient digestibility, thus improving performance [15]. The improved performance of broilers fed a diet containing 2% FO might be due to the high levels of n-3 PUFA (e.g., EPA or DHA) present in FO, that have been shown to improve both the immune status and nutrient digestibility in broilers [6]. Interestingly, low levels of dietary FO (such as 2%) have been demonstrated as being more effective than high levels of FO on improving performance [16]. The improved performance by dietary n-3 PUFA supplementation in broilers is also due to the narrowing dietary n-6/n-3 PUFA ratio (close to 4:1 or 2.5:1) and increasing the n-3 PUFA level that could effectively improve immune responses [17]. Moreover, high levels of n-3 PUFA can improve immunity and anti-inflammatory functions by decreasing the contents of inflammatory biomarkers, consequently leading to better performance in broilers [18]. The performance data in this study provides appropriate proportions as well as a reasonable mixture choice of different sources of n-3 PUFA fed to broilers.

The improved carcass traits in broilers offered the ML diet were partly in agreement with the findings of Baião et al. [19], who reported that dietary oil or fat supplementation can effectively improve the composition and quality of the carcass. The greater liver and pancreas percentages of broilers supplemented with the FS diet reflected that n-3 PUFA in FO could improve the nutrient digestibility and metabolic function [6]. The decreased relative weight of abdominal fat was mainly due to the effect of the high-level n-3 PUFA in MA, LO and FO, rather than SFA or MUFA in SO [20]. Previous studies have also concluded that LO could reduce abdominal fat deposition by promoting fatty acid β-oxidation, rather than suppressing fatty acid biosynthesis [21]. These improved body-mass percentages indicate that dietary n-3 PUFA supplementation could improve the immunity of broilers, as well as provide humans with nutritious chicken meat for consumption while reducing the instances of certain human diseases [22]. Broilers offered the FS-supplemented diet also showed higher redness values in the muscle, which may be due to the ability of FO to minimize the oxidation of myoglobin [23].

The improved concentration of serum glucose in the ML- or FS-fed broilers is due to the fatty acid profile’s (particularly PUFA/SFA ratio) ability to regulate glucose metabolism [24] and the effect of LO on lowering the level of serum insulin [25]. Broilers offered high-level n-3 PUFA diets showed lower serum TC, TG and LDL-C, which may be due to high-level n-3 PUFA regulating the contents of triacylglycerols and lipoproteins via suppressing triglycerides and apolipoprotein synthesis B [25,26]. Moreover, n-3 PUFA could also inhibit the activity of Δ9-desaturase, reducing the transformation of hepatic very low-density lipoprotein cholesterol synthesis and deterring triacylglycerol metabolism [10,27]. Furthermore, n-3 PUFA could improve Lipin-1 gene expression, which helps to control DNA-bound transcription factors to regulate gene transcription, thus regulating triglyceride synthesis [28].

Studies have demonstrated that the amount and composition of fatty acids in abdominal fat, animal tissue or muscle, as well as serum composition in broilers, are mainly modified by dietary fatty acid composition [4,26]. Therefore, the improved SFA, EPA, DHA, and n-3 PUFA deposition in the breast and thigh muscles of the FS and ML-supplemented broilers compared to the control group reflect the increased composition of these fatty acids present in the diet [10,19]. Furthermore, previous studies have shown that narrowing the n-6/n-3 PUFA ratio through the addition of FO or LO could improve performance and immune response of broilers, as well as produce n-3 PUFA-enriched chicken meat. Additionally, ALA present in LO could serve as the precursor of EPA and DHA [7]. MA has been demonstrated to improve the transfer of EPA and docosapentaenoic acid into DHA [29]. Therefore, broilers offered the ML-supplemented diet showed higher deposition of DHA. The higher concentration of DHA in broilers fed diets supplemented with FO was due to the high levels of EPA and DHA in FO that could enrich n-3 PUFA concentration in the breast and thigh muscles of poultry [3]. The decreased n-6/n-3 PUFA ratio in the breast and thigh muscles observed in this study was largely related to the dietary n-6/n-3 PUFA ratio. This is because dietary FO, LO and MA supplementation can significantly increase the n-3 PUFA and decrease the n-6 PUFA deposition in chicken muscle [30]. The beneficial nutritional recommendations in human diets of the PUFA/SFA ratio should be above 0.45, whereas the n-6/n-3 PUFA ratio should not exceed 4 and should be nearly 3:1 to 1:1. These ideal ratios could effectively optimize the bioavailability, metabolism, and incorporation of fatty acids into membrane phospholipids [31,32]. Interestingly, the PUFA/SFA ratio and n-6/n-3 PUFA ratio obtained in the breast and thigh muscles of broilers in the current study almost met the recommended guidelines, which indicates that the chicken meat harvested from broilers offered MS- or FS-supplemented diets was more nutritious for human health than broilers fed with traditional SO.

Poultry in traditional production operations may face oxidative stress and lipid oxidation, which may lead to the production of reactive oxygen species. The results in the present study demonstrated that dietary n-3 PUFA supplementation could effectively negate some of these issues, since n-3 PUFA from FO, MA and LO enhanced the antioxidant capacity of thigh and breast muscle in broilers [12]. The decreased concentration of MDA in the muscle reflected in n-3 PUFA could slow the lipid peroxidation in the muscle of broilers, while the enhanced activities of SOD and GSH-Px enzymes could work together to detoxify superoxide anions and hydrogen peroxide in cells of tissues. The increased concentration of T-AOC illustrates the improvement of the non-enzymatic antioxidant defense system in broilers [33].

The improved antioxidant status in the muscle of broilers supplemented with n-3 PUFA enriched diets was mainly due to the ability of n-3 PUFA to scavenge free radicals, H2O2 and lipid peroxides, as well as enhance the hepatic antioxidant enzymes, such as SOD and GSH-Px [34]. Moreover, n-3 PUFA could also suppress the production of pro-inflammatory cytokines (interleukin-6 and interleukin-1β) and tumor necrosis factor-α, resulting in the mitigation of increased oxidative stress and inflammatory insults while modulating lipid metabolism [35]. Furthermore, the improvement of antioxidant capacities in the muscle of broilers offered the ML-supplemented diet may help to prevent n-3 PUFA oxidation [36]. This result might also be due to the oil droplets encapsulated within the cell of MA and LO and their ability to reduce oxidative deterioration [11], as well as the effects of high-level DHA, vitamin A, vitamin E and β-carotene in MA [10,36].

## 5. Conclusions

The present research indicates that diets supplemented with 2% FO or a combination of 1% MA anday 1% LO could improve the performance, serum glucose, antioxidant capacities and n-3 PUFA profile in the muscle of broilers, as well as reduce the relative weight of abdominal fat and total serum cholesterol content of broilers compared with diets containing traditional soybean oil.

Reducing n-6/n-3 PUFA ratio via the inclusion of 2% fish oil and a mixture of 1% microalgae anday 1% linseed oil have been demonstrated as an effective method to improve antioxidant status in broilers and produce n-3 PUFA enriched chicken meat.

## Figures and Tables

**Table 1 animals-10-00508-t001:** The fatty acid compositions of linseed oil, soybean oil, fish oil and microalgae.

Fatty Acid Compositions (% Total Fatty Acids)	FO ^1^	LO ^1^	MA ^1^	SO ^1^
C14:0	2.99	0.04	5.15	0.08
C16:0	15.9	5.21	60.1	10.9
C16:1n-7	4.99	0.06	0.13	0.09
C17:0	0.66	0.06	0.56	0.10
C18:0	4.12	3.20	1.79	4.40
C18:1n-9	34.0	18.1	0.03	20.1
C18:2n-6	3.43	15.3	0.04	53.5
C18:3n-6	0.30	0.69	0.08	0.08
C18:3n-3	10.4	56.4	0.05	7.86
C20:0	1.33	0.13	0.29	0.00
C20:1n-9	0.48	0.01	0.12	0.03
C20:4n-6	0.56	0.00	0.08	0.00
C20:5n-3	3.38	0.00	0.30	0.01
C22:6n-3	4.65	0.03	28.7	0.04
SFA ^2^	25.0	8.64	67.9	15.5
MUFA ^2^	39.0	18.1	0.16	20.2
PUFA ^2^	22.7	72.4	29.3	61.5
n-6 PUFA ^2^	4.29	16.0	0.20	53.6
n-3 PUFA ^2^	18.4	56.5	29.1	7.91
n-6/n-3 PUFA ratio	0.23	0.28	0.01	6.77
PUFA/SFA ratio ^2^	0.91	8.38	0.43	3.97

^1^ FO: fish oil; SO: soybean oil; LO: linseed oil; MA: microalgae. ^2^ SFA: saturated fatty acids; MUFA: monounsaturated fatty acid; PUFA: polyunsaturated fatty acids; PUFA/SFA ratio: polyunsaturated fatty acids/saturated fatty acids ratio.

**Table 2 animals-10-00508-t002:** Ingredients and nutrient level of diets (as-fed basis, %).

Ingredients	day 1 to 21			day 22 to 42		
	CTL ^1^	ML ^1^	FS ^1^	CTL ^1^	ML ^1^	FS ^1^
Corn	58.70	58.70	58.70	65.40	65.40	65.40
Soybean meal (44%)	30.11	30.11	30.11	22.69	22.69	22.69
Corn gluten meal (62%)	4.00	4.00	4.00	5.10	5.10	5.10
Soybean oil	3.00	1.00	1.00	3.00	1.00	1.00
Microalgae	0.00	1.00	0.00	0.00	1.00	0.00
Linseed oil	0.00	1.00	0.00	0.00	1.00	0.00
Fish oil	0.00	0.00	2.00	0.00	0.00	2.00
Dicalcium phosphate	1.70	1.70	1.70	1.25	1.25	1.25
Limestone	1.39	1.39	1.39	1.44	1.44	1.44
Salt	0.30	0.30	0.30	0.30	0.30	0.30
L-lysine HCl (78%)	0.12	0.12	0.12	0.21	0.21	0.21
DL-Methionine (98%)	0.15	0.15	0.15	0.05	0.05	0.05
L-Threonine (98%)	0.03	0.03	0.03	0.06	0.06	0.06
Vitamin-mineral premix ^2^	0.50	0.50	0.50	0.50	0.50	0.50
Nutrient level ^3^						
Digestible energy, MJ/kg	12.76	12.69	12.61	13.18	13.12	13.04
Crude protein	20.70	20.64	20.65	20.10	19.94	20.00
Calcium	1.00	1.00	1.00	0.90	0.90	0.90
Digestible phosphorus	0.45	0.45	0.45	0.35	0.35	0.35
Standardized ileal digestible lysine	1.10	1.10	1.10	1.00	1.00	1.00
Standardized ileal digestible methionine	0.50	0.50	0.50	0.38	0.38	0.38
Standardized ileal digestible threonine	0.80	0.80	0.80	0.74	0.74	0.74
Standardized ileal digestible tryptophan	0.27	0.27	0.27	0.23	0.23	0.23

^1^ CTL: 3% soybean oil; ML: 1% microalgae oil + 1% linseed oil + 1% soybean oil; FS: 2% fish oil + 1% soybean oil. ^2^ Premix supplied per kg diet: vitamin A, 11,000 IU; vitamin D, 3,025 IU; vitamin E, 22 mg; vitamin K_3_, 2.2 mg; thiamine, 1.65 mg; riboflavin, 6.6 mg; pyridoxine, 3.3 mg; cobalamin, 17.6 μg; nicotinic acid, 22 mg; pantothenic acid, 13.2 mg; folic acid, 0.33 mg; biotin, 88 μg; choline chloride, 500 mg; iron, 48 mg; zinc, 96.6 mg; manganese, 101.76 mg; copper, 10 mg; selenium, 0.05 mg; iodine, 0.96 mg; cobalt, 0.3 mg. ^3^ The levels of crude protein were analyzed values, the rest were calculated values.

**Table 3 animals-10-00508-t003:** Compositions of fatty acids of the experimental diets.

Items (% of Total Fatty Acids)	day 1 to 21			day 22 to 42		
	CTL ^1^	ML ^1^	FS ^1^	CTL ^1^	ML ^1^	FS ^1^
C14:0	0.09	0.31	1.24	0.08	0.68	0.74
C16:0	17.5	18.1	17.7	16.7	18.9	14.5
C16:1n-7	0.09	0.04	0.18	0.03	0.04	1.21
C17:0	0.13	0.15	0.06	0.13	0.16	0.25
C18:0	4.38	3.06	4.09	4.11	3.05	3.10
C18:1n-9	26.3	20.9	28.7	24.2	20.2	24.8
C18:2n-6	42.2	40.2	36.0	46.0	40.9	39.1
C18:3n-6	0.00	0.00	0.10	0.00	0.00	0.00
C18:3n-3	0.12	0.07	0.15	0.10	0.07	0.12
C20:0	3.18	11.16	3.00	3.33	9.89	2.40
C20:1n-9	0.53	0.38	0.43	0.51	0.40	0.40
C20:4n-6	0.00	0.00	0.11	0.00	0.00	0.89
C20:5n-3	0.03	0.00	0.68	0.05	0.00	0.56
C22:6n-3	0.28	2.61	0.99	0.27	2.67	0.94
Saturated fatty acids	25.2	32.8	26.0	24.4	32.7	20.9
Polyunsaturated fatty acids	42.6	42.9	38.1	46.4	43.6	41.6
n-6 Polyunsaturated fatty acids	42.2	40.2	36.2	46.0	40.9	40.0
n-3 Polyunsaturated fatty acids	0.43	2.68	1.82	0.42	2.74	1.62
n-6/n-3 PUFA ratio	98.1	15.0	19.9	109	14.9	24.7
PUFA/SFA ratio ^2^	1.69	1.31	1.46	1.90	1.33	1.99

^1^ CTL: 3% soybean oil; ML: 1% microalgae oil + 1% linseed oil + 1% soybean oil; FS: 2% fish oil + 1% soybean oil. ^2^ PUFA/SFA ratio: Polyunsaturated fatty acids/Saturated fatty acids ratio.

**Table 4 animals-10-00508-t004:** Effects of different dietary fatty acids on growth performances in broilers.

Item	CTL ^1^	ML ^1^	FS ^1^	SEM	*p*-Value
BW ^2^ at day 1 (g)	44.6	46.0	45.9	0.72	0.35
BW at day 21 (g)	652 ^c^	722 ^b^	760 ^a^	11.70	<0.01
BW at day 42 (g)	2382 ^b^	2571 ^a^	2588 ^a^	32.60	<0.01
day 1 to 21					
ADG ^2^ (g)	28.9 ^c^	32.2 ^b^	34.0 ^a^	0.56	<0.01
ADFI ^2^ (g)	43.4	43.4	46.6	1.08	0.09
FCR ^2^	1.50 ^a^	1.35 ^b^	1.37 ^b^	0.03	<0.01
day 22 to 42					
ADG ^2^ (g)	82.4 ^b^	88.1 ^a^	87.0 ^a^	1.29	0.02
ADFI ^2^ (g)	141	149	136	4.35	0.14
FCR ^2^	1.72	1.69	1.56	0.05	0.13
day 1 to 42					
ADG ^2^ (g)	55.7 ^b^	60.1 ^a^	60.5 ^a^	0.77	<0.01
ADFI ^2^ (g)	92.5	96.0	91.1	2.35	0.34
FCR ^2^	1.62 ^a^	1.52 ^b^	1.47 ^b^	0.03	0.02

SEM means standard error of the mean. ^a–c^ Different superscripts within a row mean a significant difference (*p* ≤ 0.05). ^1^ CTL: 3% soybean oil; ML: 1% microalgae oil + 1% linseed oil + 1% soybean oil; FS: 2% fish oil + 1% soybean oil ^2^ ADG: Average daily gain; ADFI: Average daily feed intake; BW: Body weight; FCR: Feed conversion ratio.

**Table 5 animals-10-00508-t005:** Effects of different dietary fatty acids on carcass traits of broilers.

Item	CTL ^1^	ML ^1^	FS ^1^	SEM	*p*-Value
day 21					
Breast muscle					
Lightness	42.8	44.1	42.3	2.36	0.85
Redness	6.56	5.75	5.62	0.58	0.49
Yellowness	8.88	10.0	9.39	1.01	0.74
Thigh muscle					
Lightness	50.7 ^ab^	51.8 ^a^	47.6 ^b^	1.08	0.05
Redness	6.09	6.24	7.43	1.00	0.59
Yellowness	10.4	11.2	9.71	0.76	0.42
d 42					
Breast muscle					
Lightness	38.2	37.2	37.9	0.90	0.76
Redness	3.11 ^b^	3.62 ^a,b^	4.31 ^a^	0.31	0.05
Yellowness	8.84	8.66	9.34	0.62	0.74
Drop loss (%)	2.14	1.67	1.63	0.68	0.74
Thigh muscle					
Lightness	39.2	37.8	38.0	1.30	0.71
Redness	4.31 ^b^	4.07 ^b^	6.34 ^a^	0.48	0.01
Yellowness	9.93	9.88	10.2	0.57	0.92
Drop loss (%)	0.83	0.96	1.03	0.06	0.16
Relative weight of organs (% BW)					
Heart	0.58	0.56	0.61	0.05	0.81
Liver	1.98 ^b^	2.06 ^b^	2.52 ^a^	0.12	0.01
Spleen	1.07	1.30	1.26	0.18	0.42
Pancreas	0.17 ^b^	0.18 ^a,b^	0.21 ^a^	0.01	0.02
Abdominal fat	0.20 ^a^	0.16 ^b^	0.14 ^b^	0.01	0.01
Fabricius	0.26	0.22	0.24	0.03	0.53

SEM means standard error of the mean. ^a^^,b^ Different superscripts within a row indicate a significant difference (*p* ≤ 0.05). ^1^ CTL: 3% soybean oil; ML: 1% microalgae oil + 1% linseed oil + 1% soybean oil; FS: 2% fish oil + 1% soybean oil.

**Table 6 animals-10-00508-t006:** Effects of different dietary fatty acids on serum composition of broilers.

Item	CTL ^1^	ML ^1^	FS ^1^	SEM	*p*-Value
day 1 to 21					
Glucose (mmol/L)	17.3	19.0	19.4	0.66	0.12
Albumin (g/L)	12.3	12.4	13.6	0.52	0.25
Globulin (g/L)	19.1	16.9	18.3	0.90	0.28
Total protein (g/L)	31.5	29.3	31.9	1.54	0.33
Urea nitrogen (mmol/L)	0.46	0.62	0.53	0.09	0.39
Total cholesterol (mmol/L)	4.57 ^a^	3.33 ^b^	3.39 ^b^	0.18	<0.01
Triglyceride (mmol/L)	0.77	0.58	0.60	0.17	0.73
LDL-C ^2^ (mmol/L)	0.52	0.51	0.55	0.03	0.39
HDL-C ^2^ (mmol/L)	1.47	1.32	1.44	0.06	0.19
day 22 to 42					
Glucose (mmol/L)	14.5 ^b^	18.3 ^a^	18.0 ^a^	0.83	0.03
Albumin (g/L)	12.3	11.4	12.6	0.62	0.17
Globulin (g/L)	17.6	17.4	20.8	1.82	0.27
Total protein (g/L)	29.9	28.8	33.4	2.37	0.24
Urea nitrogen (mmol/L)	0.35	0.20	0.18	0.05	0.09
Total cholesterol (mmol/L)	3.30	2.78	3.08	0.12	0.10
Triglyceride (mmol/L)	0.77	0.60	0.62	0.07	0.29
LDL-C ^2^ (mmol/L)	0.42	0.39	0.30	0.04	0.15
HDL-C ^2^ (mmol/L)	1.39	1.27	1.33	0.06	0.44

SEM means standard error of the mean. ^a,b^ Different superscripts within a row indicate a significant difference (*p* ≤ 0.05). ^1^ CTL: 3% soybean oil; ML: 1% microalgae oil + 1% linseed oil + 1% soybean oil; FS: 2% fish oil + 1% soybean oil. ^2)^ LDL-C: low-density lipoprotein cholesterol; HDL-C: high-density lipoprotein cholesterol.

**Table 7 animals-10-00508-t007:** Effects of dietary fatty acids on antioxidant capacity in breast and thigh muscle of broilers.

Item	CTL ^1^	ML ^1^	FS ^1^	SEM	*p*-Value
Breast muscle					
Superoxide dismutase, U/mg	114 ^b^	214 ^a^	209 ^a^	8.69	<0.01
Total antioxidant capacity, U/mg	5.63 ^c^	12.3 ^a^	9.81 ^b^	0.34	<0.01
Glutathione peroxidase, U/mg	699 ^b^	1281 ^a^	1131 ^a^	52.8	<0.01
Catalase, U/mg	5.26	9.28	9.37	1.31	0.15
Malondialdehyde, nmol/mg	5.64 ^a^	2.95 ^b^	3.16 ^b^	0.34	<0.01
Thigh muscle					
Superoxide dismutase, U/mg	118 ^b^	240 ^a^	210 ^a^	10.1	<0.01
Total antioxidant capacity, U/mg	5.71 ^c^	14.1 ^a^	9.86 ^b^	0.96	<0.01
Glutathione peroxidase, U/mg	661 ^c^	1324 ^a^	927 ^b^	32.3	<0.01
Catalase, U/mg	6.36 ^b^	12.17 ^a^	8.22 ^b^	0.77	0.01
Malondialdehyde, nmol/mg	4.67 ^a^	2.75 ^b^	3.22 ^b^	0.27	0.02

SEM means standard error of the mean. ^a–c^ Different superscripts within a row indicate a significant difference (*p* ≤ 0.05). ^1^ CON: 3% soybean oil, in the control group; ML: 1% microalgae oil + 1% linseed oil + 1% soybean oil; FS: 2% fish oil + 1% soybean oil.

**Table 8 animals-10-00508-t008:** Effects of different dietary fatty acids on breast muscle fatty acid composition of broilers (g/100 g total fatty acids).

Item	CTL ^1^	ML ^1^	FS ^1^	SEM	*p*-Value
C14:0	0.47 ^c^	0.59 ^b^	0.70 ^a^	0.02	<0.01
C16:0	21.8	23.0	23.4	0.67	0.31
C16:1n-7	4.18	3.90	4.45	0.34	0.56
C17:0	0.12 ^b^	0.14 ^a^	0.15 ^a^	0.01	0.01
C18:0	8.40	8.54	9.03	0.23	0.24
C18:1n-9	31.9	30.1	33.5	0.84	0.10
C18:2n-6	25.6 ^a^	19.3 ^b^	19.3 ^b^	0.51	<0.01
C18:3n-6	0.28	0.22	0.23	0.02	0.25
C18:3n-3	2.02 ^b^	4.34 ^a^	1.26 ^c^	0.11	<0.01
C20:0	0.12	0.12	0.12	0.01	0.43
C20:1n-9	0.25 ^b^	0.28 ^b^	0.57 ^a^	0.02	<0.01
C20:4n-6	2.46	2.52	2.40	0.33	0.97
C20:5n-3	0.12 ^b^	0.68 ^a^	0.56 ^a^	0.06	<0.01
C22:6n-3	0.30 ^c^	3.46 ^a^	1.72 ^b^	0.24	<0.01
SFA	30.9	32.4	33.4	0.79	0.19
MUFA	36.1	34.0	38.0	0.81	0.06
PUFA	30.8 ^a^	30.5 ^a^	25.4 ^b^	0.85	0.02
n-6 PUFA	28.4 ^a^	22.0 ^b^	21.9 ^b^	0.68	<0.01
n-3 PUFA	2.44 ^c^	8.49 ^a^	3.53 ^b^	0.25	<0.01
n-6/n-3 PUFA ratio	11.7 ^a^	2.59 ^c^	6.35 ^b^	0.45	<0.01
PUFA/SFA ratio ^2^	1.00 ^a^	0.94 ^a^	0.77 ^b^	0.04	0.04

SEM means standard error of the mean. ^a–c^ Different superscripts within a row indicate a significant difference (*p* ≤ 0.05). ^1^ CTL: 3% soybean oil; ML: 1% microalgae oil + 1% linseed oil + 1% soybean oil; FS: 2% fish oil + 1% soybean oil. ^2^ PUFA/SFA ratio: Polyunsaturated fatty acids/Saturated fatty acids ratio.

**Table 9 animals-10-00508-t009:** Effects of dietary fatty acids on thigh muscle fatty acid composition of broilers (g/100 g total fatty acids).

Item	CTL ^1^	ML ^1^	FS ^1^	SEM	*p*-Value
C14:0	0.47 ^c^	0.54 ^b^	0.66 ^a^	0.01	<0.01
C16:0	21.7	23.1	22.9	0.37	0.11
C16:1n-7	3.88	3.34	3.11	0.20	0.11
C17:0	0.12 ^c^	0.15 ^b^	0.17 ^a^	0.01	<0.01
C18:0	8.13 ^b^	9.53 ^b^	11.1 ^a^	0.36	0.01
C18:1n-9	32.6 ^a^	28.9 ^b^	27.9 ^b^	0.60	0.01
C18:2n-6	25.7 ^a^	17.7 ^b^	18.9 ^b^	0.49	<0.01
C18:3n-6	0.33 ^a^	0.18 ^b^	0.16 ^b^	0.03	0.03
C18:3n-3	2.25 ^b^	3.37 ^a^	1.14 ^c^	0.09	<0.01
C20:0	0.18	0.15	0.19	0.02	0.56
C20:1n-9	0.23	0.28	0.50	0.06	0.08
C20:4n-6	2.18 ^b^	3.08 ^b^	4.45 ^a^	0.26	0.01
C20:5n-3	0.17 ^b^	0.76 ^a^	0.85 ^a^	0.06	<0.01
C22:6n-3	0.23 ^b^	5.22 ^a^	3.60 ^a^	0.43	<0.01
SFA	30.6 ^b^	33.5 ^a^	35.0 ^a^	0.48	<0.01
MUFA	36.4 ^a^	32.3 ^b^	31.0 ^b^	0.75	0.02
PUFA	30.8	30.3	29.0	0.74	0.31
n-6 PUFA	28.2 ^a^	21.0 ^b^	23.5 ^b^	0.68	<0.01
n-3 PUFA	2.65 ^c^	9.35 ^a^	5.58 ^b^	0.46	<0.01
n-6/n-3 PUFA ratio	10.7 ^a^	2.26 ^c^	4.25 ^b^	0.42	<0.01
PUFA/SFA ratio ^2^	1.01 ^a^	0.91 ^a^^,^^b^	0.83 ^b^	0.03	0.04

SEM means standard error of the mean. ^a–c^ Different superscripts within a row indicate a significant difference (*p* ≤ 0.05). ^1^ CTL: 3% soybean oil; ML: 1% microalgae oil + 1% linseed oil + 1% soybean oil; FS: 2% fish oil + 1% soybean oil. ^2^ PUFA/SFA ratio: Polyunsaturated fatty acids/Saturated fatty acids ratio.

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
