# Peer review of "Effects of Dietary Fatty Acids from Different Sources on Growth Performance, Meat Quality, Muscle Fatty Acid Deposition, and Antioxidant Capacity in Broilers"

_animals, 2020, doi:10.3390/ani10030508_

Round 1

Reviewer 1 Report

This referee has some major concerns regarding this submission.

First, this manuscript is readable, but suffers from excessive number of minor errors in language, spelling and presentation style, and will require major proof-reading.  It is not possible for this reviewer to list any of these issues. It is recognised that English is probably not the first language of the authors, but the authors are responsible (or to seek the services of English editorial advice) to improve the clarity that will meet the standards required for publication in an international journal.

Second, there is nothing novel about the reported findings. Efficient transfer of fatty acids from the feed to broiler meat is well established and the results are as expected. Essentially the findings add little to our current knowledge on this topic.

Also, authors seem to promote this meat enrichment in terms of human nutrition. Note that for humans to benefit, one has to eat kilos of enriched meat every day. The focus should be on broilers. Therefore, statements need to be modifies: L12-16   - redundant. L43-57 – must be modified.

The basis for the combinations used is not clear.

L95:     Why NRC (1994). Over 25 years old and not relevant to current broiler strains.

Table 2: Why GE? – not relevant. How dig P calculated. What was the source?

L154    Duncan’s is a weak test and must be avoided.

Table 5: Relative weight of organs (%BW) – not organ %. Change here and throughout.

L239: There is no growth-promoting effect – rephase

L239: MS??

L255: this reasoning is incorrect – Final BW will not improve these.

L317: is this economically viable? – feeding FO or MA for 42d? – NO!!

Author Response

Revision Note, (List of modification) Date: 2020-03-06 (y-m-d)

Manuscript ID: animals-737018-RI.

Dear Sir

Good day. Thank you very much for your kind consideration with our submitted article and offering us the further opportunity to submit the revised manuscript. Please find here the point to point expert reviewer’s and editor’s comments with necessary changes as per suggested with this attached file, and the amendments are highlighted in red in the revised manuscript. We have revised our manuscript for language and grammar checked by a native English speaker working in our University. We do thanks to skilled reviewers, academic editors and editorial board members as well for their critical evaluation to make the manuscript more effective for review process in Animals Journal.

Many thanks.

Sincerely yours,

Prof. Dr. Xiang Shu Piao,

State Key laboratory of Animal Nutrition, College of Animal Science and Technology, China Agricultural University, Beijing 100193, China

Corresponding Author,

Email: piaoxsh@cau.edu.cn; Tel.:+86-1062733588/Fax.: +86-1062733688

Reviewer 1

This referee has some major concerns regarding this submission.

Comment: First, this manuscript is readable, but suffers from excessive number of minor errors in language, spelling and presentation style, and will require major proof-reading. It is not possible for this reviewer to list any of these issues. It is recognized that English is probably not the first language of the authors, but the authors are responsible (or to seek the services of English editorial advice) to improve the clarity that will meet the standards required for publication in an international journal.

Response: Thanks for the advice, we have revised our manuscript for language and grammar checked by a native English speaker working in our University.

Comment: Second, there is nothing novel about the reported findings. Efficient transfer of fatty acids from the feed to broiler meat is well established and the results are as expected. Essentially the findings add little to our current knowledge on this topic. Also, authors seem to promote this meat enrichment in terms of human nutrition. Note that for humans to benefit, one has to eat kilos of enriched meat every day. The focus should be on broilers. Therefore, statements need to be modifies.

Response: Thanks for the advice, in this study, we give some choices for the use of different resources of fatty acid inbroilers, and the use of a combination of microalgae and linseed oil is few reported in the previous study, which could be a novel combination choice. Although the results met our expectation, this study also give us some information for the choice of oil supplementation in broilers’ diet, and the impacts of them on performance carcass traits, meat quality, meat fatty acid deposition, and antioxidant capacity in broilers compared with traditional soybean oil, which could be an important topic in chicken nutrition science.

Besides, we have changed some statements and focused more on dietary fatty acids supplementation on the growth and health status of broilers in introduction and conclusion parts.

Comment: L12-16- redundant.

Response: Thanks for that recommendation. I have delete L12-16.

Comment: L43-57 – must be modified.

Response: Thanks for that recommendation. I have modified the statement in L 43-57 to make the sentence more reasonable.

Comment: The basis for the combinations used is not clear.

Response: Thanks for that recommendation. I have better organize the basis for the combinations. Microalgae is rich with DHA (could improve antioxidant status in bird), linseed oil is rich in ALA (could modulate fatty acid profile), while FO is common used n-3 PUFA, and SO is mainly used fatty acid in broilers. Therefore, based on previous findings, we hypothesis that dietary MA and LO supplementation might be a novel combination for improving health status and muscle fatty acid profile in birds in comparison with FO or traditional SO.

Comment: L95: Why NRC (1994). Over 25 years old and not relevant to current broiler strains.

Response: Thanks for that recommendation. We also refer to the AA broiler nutrients recommendation in 2019. There are some published papers showing the reference NRC (1994) may be also suitable to use. And the growth performance using this standard is reasonable by this diet according to the current study.

Comment: Table 2: Why GE? – not relevant. How dig P calculated. What was the source?

Response: Thanks for that recommendation. We have added DE, and Dig P, which was calculated by the sum of calculated Dig P content in all the used ingredients in this study.

Comment: L154 Duncan’s is a weak test and must be avoided.

Response: Thanks for that recommendation. We have used other data analysis, the Student-Neuman-Keul’s Multiple Range Test.

Comment: Table 5: Relative weight of organs (%BW) – not organ %. Change here and throughout.

Response: Thanks for that recommendation. We have changed this part.

Comment: L239: There is no growth-promoting effect – rephrase

Response: Thanks for that recommendation. We found there is an ADG improvement in birds fed ML compared with control.

Comment: L241: MS??

Response: Thanks for that recommendation. We have corrected this mistakes. Comment: L255: this reasoning is incorrect – Final BW will not improve these.

Response: Thanks for that recommendation. We have reorganized the language and better explained it.

Comment: L317: is this economically viable? – feeding FO or MA for 42d? – NO!!

Response: Thanks for the comments. We have avoided using such kind of words.

Thanks

Reviewer 2 Report

The idea is good and the paper contains useful information, the paper merits the acceptance after minor revision as below:

Introduction needs more improvements

In table 2: why the authors used the same CP % in the starter and finisher phases? I think the starter CP is at least 23% (recommended level)

Line 156: Significance should be significance

Line 317-320: use the abbreviations in the conclusion (fish oil or a combination of 1% microalgae 317 and 1% linseed oil)

Author Response

Revision Note, (List of modification) Date: 2020-03-06 (y-m-d)

Manuscript ID: animals-737018-RI.

Dear Sir

Good day. Thank you very much for your kind consideration with our submitted article and offering us the further opportunity to submit the revised manuscript. Please find here the point to point expert reviewer’s and editor’s comments with necessary changes as per suggested with this attached file, and the amendments are highlighted in red in the revised manuscript. We have revised our manuscript for language and grammar checked by a native English speaker working in our University. We do thanks to skilled reviewers, academic editors and editorial board members as well for their critical evaluation to make the manuscript more effective for review process in Animals Journal.

Many thanks.

Sincerely yours,

Prof. Dr. Xiang Shu Piao,

State Key laboratory of Animal Nutrition, College of Animal Science and Technology, China Agricultural University, Beijing 100193, China

Corresponding Author,

Email: piaoxsh@cau.edu.cn; Tel.:+86-1062733588/Fax.: +86-1062733688

Reviewer 2:

Comment: The idea is good and the paper contains useful information, the paper merits the acceptance after minor revision as below:

Response: Thanks for the comment, we have corrected all the mistakes according to your useful advice.

Comment: Introduction needs more improvements

Response: Thanks for that recommendation. We have improved our introduction part.

Comment: In table 2: why the authors used the same CP % in the starter and finisher phases? I think the starter CP is at least 23% (recommended level)

Response: Thanks for that recommendation. We have rechecked the data, and we found the amino acids levels met or exceed the requirement of NRC (1994). And we found the growth performance met the standard of AA broilers.

Comment: Line 156: Significance should be significance

Response: Thanks for that recommendation. We have corrected this mistake.

Comment: Line 317-320: use the abbreviations in the conclusion (fish oil or a combination of 1% microalgae 317 and 1% linseed oil)

Response: Thanks for that recommendation. We have used the abbreviations in the conclusion part.

Thanks

Reviewer 3 Report

The authors investigated the impacts of the different resources of fatty acid in diet on the broiler performance and meat quality. They revealed that dietary fish oil or a combination of linseed oil and microalgae could be effective ways on improving growth performance, carcass traits, meat fatty acid deposition, and antioxidant capacity in broilers compared with traditional soybean oil in broilers. This topic is important in chicken nutrition science, however the methodology chapter is not enough to explain how the authors reached these results. So my comment is essential in developing methodology including scientific details for each experiment to become more clear for others. I found it difficult to review a lot of abbreviations and it was difficult for me to investigate, so the authors recommended a strict review of each one in the manuscript. Also, I consider the manuscript to be linguistically clear, using direct sentences and a few simple mistakes. Herein, I listed some comments to be consider before publish the manuscript.

L35: replace “breast and thigh meat” to breast and thigh muscle and in throughout manuscript

L115: replace Broilers to “Birds” and in throughout manuscript

L117: it should be mentioned for individuals using “bird or chick” not use broiler.

L122: The protocols used in serum biochemical parameters measurement should be briefly described.

L127-131: The protocols used in antioxidant enzymes measurement should be briefly described

L123-131: Check carefully which sample and number used in each experiments (biochemical parameters and enzymes)

L133: why the author measured meat quality at d 21 to, still not complete growth and composition.

L133: what is specific part of muscle taken to determine meat quality, different muscle leads to different results?

L135: please provide the equations for calculation of color chroma, I suggest to mention this reference “doi:10.3390/ani9121134”

L139: replace organ percentages to “organ indexes” and in throughout manuscript

L143: Again which part in breast and thigh muscle tacked to measure fatty profile? It is important to be sure same anatomy part already collected to adjust the results.

L151: please provide statistical model component and experimental unit

L233: Fig 1 need to improve or change to table

L278: I recommend use this reference doi:10.1017/S1751731119000508

L303: pleas use concentration with MDA because its marker for lipid peroxidation and not enzyme, while use activity or activities not concentration with other enzymes such as SOD and GSH-Px in throughout the manuscript

L307-321: The summary chapter needs to be reformulated according to the results

Author Response

Revision Note, (List of modification) Date: 2020-03-06 (y-m-d)

Manuscript ID: animals-737018-RI.

Dear Sir

Good day. Thank you very much for your kind consideration with our submitted article and offering us the further opportunity to submit the revised manuscript. Please find here the point to point expert reviewer’s and editor’s comments with necessary changes as per suggested with this attached file, and the amendments are highlighted in red in the revised manuscript. We have revised our manuscript for language and grammar checked by a native English speaker working in our University. We do thanks to skilled reviewers, academic editors and editorial board members as well for their critical evaluation to make the manuscript more effective for review process in Animals Journal.

Many thanks.

Sincerely yours,

Prof. Dr. Xiang Shu Piao,

State Key laboratory of Animal Nutrition, College of Animal Science and Technology, China Agricultural University, Beijing 100193, China

Corresponding Author,

Email: piaoxsh@cau.edu.cn; Tel.:+86-1062733588/Fax.: +86-1062733688

Reviewer 3:

Main comments: The authors investigated the impacts of the different resources of fatty acid in diet on the broiler performance and meat quality. They revealed that dietary fish oil or a combination of linseed oil and microalgae could be effective ways on improving growth performance, carcass traits, meat fatty acid deposition, and antioxidant capacity in broilers compared with traditional soybean oil in broilers. This topic is important in chicken nutrition science, however the methodology chapter is not enough to explain how the authors reached these results. So my comment is essential in developing methodology including scientific details for each experiment to become more clear for others. I found it difficult to review a lot of abbreviations and it was difficult for me to investigate, so the authors recommended a strict review of each one in the manuscript. Also, I consider the manuscript to be linguistically clear, using direct sentences and a few simple mistakes. Herein, I listed some comments to be consider before publish the manuscript.

Main response: Thanks for that recommendation, we have revised the methodology chapter part in manuscript according to the advice. We have reviewed abbreviations used in this manuscript and revised our manuscript for language and grammar checked by a native English speaker working in our University.

Comment: L35: replace “breast and thigh meat” to breast and thigh muscle and in throughout manuscript

Response: Thanks for the suggestion! We have changed all these parts in this paper.

Comment: L115: replace Broilers to “Birds” and in throughout manuscript

Response: Thanks for that recommendation. We have changed all these parts in this paper.

Comment: L117: it should be mentioned for individuals using “bird or chick” not use broiler.

Response: Thanks for that recommendation. We have changed this parts in the revised manuscript.

Comment: L122: The protocols used in serum biochemical parameters measurement should be briefly described.

Response: Thanks for that recommendation. We have added those parts in the M&M

Comment: L127-131: The protocols used in antioxidant enzymes measurement should be briefly described.

Response: Thanks for that recommendation. We have added those parts in the M&M. The concentration of malondialdehyde (MDA) was determined by Thiobarbituric Acid Reactive Substance (TBARS) Assay Kit (Cell Biolabs, Inc., San Diego, CA) following the manufacturer’s protocols. The activities of superoxide dismutase (SOD), catalase (CAT), total antioxidant capacity (T-AOC), and glutathione peroxidase (GSH-Px) in serum were determined by spectrophotometric methods using a spectrophotometer (LengGuang SFZ1606017568), according to the SOD, CAT, T-AOC and GSH-Px kits’ manufacturer instructions (The Nanjing Jiancheng Bioengineering Institute, Nanjing, China).

Comment: L123-131: Check carefully which sample and number used in each experiments (biochemical parameters and enzymes)

Response: Thanks for that recommendation. We have added those parts in the M&M. We use the serum samples in bird of each pen (n = 7) to measure the biochemical parameters and enzymes activities.

Comment: L133: why the author measured meat quality at d 21 to, still not complete growth and composition.

Response: Thanks for that recommendation. We have reconsidered those parts. We want to review the effects of different fatty acids supplementation on the development of body in birds.

Comment: L133: what is specific part of muscle taken to determine meat quality, different muscle leads to different results?

Response: Thanks for that recommendation. We have added those parts in the M&M. The muscle color (lightness, redness, and yellowness values) in the left breast muscle and thigh muscle were measured from 3 orientations (middle, medial, and lateral) using a Chromameter (CR-410, Konica Minolta, Tokyo, Japan).

Comment: L135: please provide the equations for calculation of color chroma, I suggest to mention this reference “doi:10.3390/ani9121134”

Response: Thanks for the comments! In the current study, we did not calculate the color chroma, but we learn the method from the reference “doi:10.3390/ani9121134”, so we have added this reference part in the M&M

Comment: L139: replace organ percentages to “organ indexes” and in throughout manuscript

Response: Thanks for that recommendation. We have replaced organ percentages to “organ indexes” and in throughout manuscript

Comment: L143: Again which part in breast and thigh muscle tacked to measure fatty profile? It is important to be sure same anatomy part already collected to adjust the results.

Response: Thanks for that recommendation. We have added those parts in the M&M. And we used the left breast and left thigh muscle to measure fatty profile.

Comment: L151: please provide statistical model component and experimental unit

Response: Thanks for that recommendation. We have added those parts in the data analysis part. All data were analyzed using the MIXED procedure of SAS version 9.1. A cage was used as the experimental unit for growth performance (body weight gain, ADG, ADFI and FCR); individual bird was considered the experimental unit for all other indices.

Therefore, our expression changed into: Data were analyzed using the MIXED procedures of SAS (version 9.2, 2008; Inst., Inc., Cary, NC). The dietary treatment served as the fixed effect, and body weight was the random effect. The individual pen was classified as the experimental unit in terms of performance data (body weight gain, ADG, ADFI and FCR). Individual bird selected in each pen was considered as the experimental unit for all other indices. Student-Neuman-Keul’s Multiple Range Test was used to separate the statistical differences among all the treatments. Least squares means and SEM were used to express results. The significance was defined at P ≤ 0.05, and a trend for significance was determined at 0.05 < P ≤ 0.10.

Comment: L233: Fig 1 need to improve or change to table

Response: Thanks for that recommendation. We have changed it into Table

Comment: L278: I recommend use this reference doi:10.1017/S1751731119000508

Response: Thanks for that recommendation. Since essential oil in the reference of doi:10.1017/S1751731119000508 is not fatty acid resource, we may not cite this reference in revised manuscript, but we used other expression to make the discuss better.

Comment: L303: please use concentration with MDA because its marker for lipid peroxidation and not enzyme, while use activity or activities not concentration with other enzymes such as SOD and GSH-Px in throughout the manuscript

Response: Thanks for that recommendation. We have referred to these recommendations in this manuscript.

Comment: L307-321: The summary chapter needs to be reformulated according to the results

Response: Thanks for that recommendation. I have rewrite the summary chapter. The conclusion part was changed into: The present research indicates that diets supplemented with 2% FO or a combination of 1% MA and 1% LO could improve performance, serum glucose, antioxidant capacities and n-3 PUFA profile in the muscle of birds, as well as reduce the abdominal fat index and total serum cholesterol content of birds compared with diets containing traditional soybean oil. Reducing n-6/n-3 PUFA ratio via the inclusion of 2% fish oil and a mixture of 1% microalgae and 1% linseed oil have been demonstrated as an effective method to improve antioxidant status in birds and produce n-3 PUFA enriched chicken meat.

Thanks!

Round 2

Reviewer 1 Report

Still there are excessive number of minor errors in language and grammar throughout the manuscript. Careful proof-reading by a native English editor is recommended

A particular concern is the use of term 'birds' instead of 'broilers'. 

Some other issues:

L2  sources, not resources

L4  broilers, not birds

L18  pens or cages? 

L24 relative weights, not indexes

L97  not clear

Author Response

Revision Note, (List of modification) Date: 2020-03-13 (y-m-d)

Manuscript ID: animals-737018-R2.

Dear Sir

Good day. Thank you very much for your kind consideration with our submitted article and offering us the further opportunity to submit the revised manuscript. Please find here the point to point expert reviewer’s and editor’s comments with necessary changes as per suggested with this attached file, and the amendments are highlighted in red in the revised manuscript. We have revised our manuscript for language and grammar checked by a native English speaker working in our University. We do thanks to skilled reviewers, academic editors and editorial board members as well for their critical evaluation to make the manuscript more effective for review process in Animals Journal.

Many thanks.

Sincerely yours,

Prof. Dr. Xiang Shu Piao,

State Key laboratory of Animal Nutrition, College of Animal Science and Technology, China Agricultural University, Beijing 100193, China

Corresponding Author,

Email: piaoxsh@cau.edu.cn; Tel.:+86-1062733588/Fax.: +86-1062733688
Comment
: Still there are excessive number of minor errors in language and grammar throughout the manuscript. Careful proof-reading by a native English editor is recommended.

Response: We have corrected the language and grammar errors in this manuscript by a native English speaker working in our University

Comment: A particular concern is the use of term 'birds' instead of 'broilers'.

Response: We have changed the term “birds” into “broilers”.

Comment: L2 sources, not resources

Response: We have changed this error. We have changed the title into “Effects of dietary fatty acids from different sources on growth performance, meat quality, muscle fatty acid deposition, and antioxidant capacity in broilers”.

Comment: L4 broilers, not birds

Response: We have corrected this part in this manuscript

Comment: L18 pens or cages?

Response: We have corrected this part in this manuscript

Comment: L24 relative weights, not indexes

Response: We have corrected this part in this manuscript

Comment: L97 not clear

Response: We have changed “Nutrient levels in all formulated diets for each phase met or exceeded the nutrient requirements recommended in NRC (1994, Table 2), and the experimental diets were formulated on the basis of a previous study by Long et al. [10]” into “Nutrient levels in all formulated diets for each phase met or exceeded the nutrient requirements recommended in NRC (1994, Table 2)”.

Thanks!

Reviewer 3 Report

I accept publish the revised version

Author Response

Revision Note, (List of modification) Date: 2020-03-13 (y-m-d)

Manuscript ID: animals-737018-R2.

Dear Sir

Good day. Thank you very much for your kind consideration with our submitted article and offering us the further opportunity to submit the revised manuscript. Please find here the point to point expert reviewer’s and editor’s comments with necessary changes as per suggested with this attached file, and the amendments are highlighted in red in the revised manuscript. We have revised our manuscript for language and grammar checked by a native English speaker working in our University. We do thanks to skilled reviewers, academic editors and editorial board members as well for their critical evaluation to make the manuscript more effective for review process in Animals Journal.

Many thanks.

Sincerely yours,

Prof. Dr. Xiang Shu Piao,

State Key laboratory of Animal Nutrition, College of Animal Science and Technology, China Agricultural University, Beijing 100193, China

Corresponding Author,

Email: piaoxsh@cau.edu.cn; Tel.:+86-1062733588/Fax.: +86-1062733688

Comments and Suggestions for Authors: I accept publish the revised version

Response: Thanks for your sincere help!
